# Natural Compounds and Biomimetic Engineering to Influence Fibroblast Behavior in Wound Healing

**DOI:** 10.3390/ijms25063274

**Published:** 2024-03-14

**Authors:** Charlotte E. Berry, Camille Brenac, Caroline E. Gonzalez, Carter B. Kendig, Thalia Le, Nicholas An, Michelle F. Griffin

**Affiliations:** 1Hagey Laboratory for Pediatric Regenerative Medicine, Division of Plastic and Reconstructive Surgery, Department of Surgery, Stanford University School of Medicine, 257 Campus Drive West, Stanford, CA 94305, USA; berryc@stanford.edu (C.E.B.); c270795@stanford.edu (C.B.); carter.kendig@einsteinmed.edu (C.B.K.); thaliale0711@gmail.com (T.L.); nicholas.j.an.med@dartmouth.edu (N.A.); 2Feinberg School of Medicine, Northwestern University, Chicago, IL 60611, USA; caroline.gonzalez@northwestern.edu

**Keywords:** natural compounds, biomimetic engineering, wound healing, fibroblast, fibroblast behavior

## Abstract

Throughout history, natural products have played a significant role in wound healing. Fibroblasts, acting as primary cellular mediators in skin wound healing, exhibit behavioral responses to natural compounds that can enhance the wound healing process. Identifying bioactive natural compounds and understanding their impact on fibroblast behavior offers crucial translational opportunities in the realm of wound healing. Modern scientific techniques have enabled a detailed understanding of how naturally derived compounds modulate wound healing by influencing fibroblast behavior. Specific compounds known for their wound healing properties have been identified. Engineered biomimetic compounds replicating the natural wound microenvironment are designed to facilitate normal healing. Advanced delivery methods operating at micro- and nano-scales have been developed to effectively deliver these novel compounds through the stratum corneum. This review provides a comprehensive summary of the efficacy of natural compounds in influencing fibroblast behavior for promoting wound regeneration and repair. Additionally, it explores biomimetic engineering, where researchers draw inspiration from nature to create materials and devices mimicking physiological cues crucial for effective wound healing. The review concludes by describing novel delivery mechanisms aimed at enhancing the bioavailability of natural compounds. Innovative future strategies involve exploring fibroblast-influencing pathways, responsive biomaterials, smart dressings with real-time monitoring, and applications of stem cells. However, translating these findings to clinical settings faces challenges such as the limited validation of biomaterials in large animal models and logistical obstacles in industrial production. The integration of ancient remedies with modern approaches holds promise for achieving effective and scar-free wound healing.

## 1. Introduction

Wound healing comprises three distinct yet interconnected stages: inflammation, proliferation, and remodeling [1]. While the ultimate goal of wound repair is functional skin regeneration, in adults, the process often results in the formation of nonfunctional fibrotic tissue, commonly referred to as “scars” [2,3]. Pathological wound healing can manifest as underhealing (delayed repair) or overhealing. Excessive wound healing leads to hypertrophic/keloid scars, causing patients significant physical and psychological morbidity [1]. Cellular and molecular pathways, particularly those involving fibroblasts, are instrumental in mediating wound healing and scar formation.

Fibroblasts play a crucial role in scar formation and fibrosis, making them a focal point in research aimed at improving wound healing outcomes [4]. These cells, abundantly present in wounds, coordinate the healing process at every stage by producing regulatory molecules and recruiting immune cells (Figure 1) [5]. Therefore, the dysfunction of fibroblasts could lead to delayed wound closure or excessive fibrosis. Initially, wound healing involves coagulation and inflammation in response to a damaged extracellular matrix (ECM) and damaged resident cells. This first inflammatory phase includes the vasoconstriction of injured vessels and a complex sequence of reactions in response to blood contact, including plasma factor VII/VIIa binding to tissue factor 3, ultimately producing a fibrin and platelet clot [6].

Platelets, in addition to stimulating the coagulation cascade, secrete growth factors such as transforming growth factor-β (TGF-β) and platelet-derived growth factor (PDGF), activating the recruitment of fibroblasts and immune cells [5]. Platelets play an important role in enhancing blood vessel permeability through the secretion of histamine and serotonin, alongside nitric oxide production by endothelial cells, which facilitates vasodilation. Concurrently, platelet-derived signals, complement activation, and factors from the coagulation cascade trigger chemotaxis, prompting immune cell migration to the wound bed for the elimination of potential pathogens or foreign debris. Although neutrophils are not deemed necessary for wound healing, they generate inflammatory products vital for wound repair and undertake the phagocytosis of pathogens. Conversely, both skin-resident and monocyte-derived macrophages are pivotal for wound healing, emerging as the dominant cell population orchestrating the release of growth factors within the wound environment until fibroblast proliferation [7]. Fibroblasts, typically arriving at the wound site between days 5 and 7, alongside lymphocytes, release inflammatory cytokines and chemokines, thereby regulating the wound microenvironment [8].

During the proliferation phase, fibroblasts play a pivotal role in wound remodeling by secreting matrix metalloproteinases (MMPs) to break down the fibrin clot [9]. Fibroblast growth factor (FGF) and TGF-β induce fibroblast proliferation and granulation tissue formation, facilitating the production of ECM molecules for keratinocyte migration [10]. In the remodeling phase, fibroblasts remodel the ECM and differentiate into myofibroblasts, regulating mechanical wound contraction [11]. This transformation is governed by cytokines such as TGF-β1 and CXC motif chemokine ligand 8, along with ECM stiffness. Myofibroblasts, acquiring contractile properties, enhance the organization and tensile strength of the ECM. As the final stage of wound healing, the remodeling phase can last from weeks to years [4].

Given the fundamental role of fibroblasts in normal tissue repair, dysregulation in their behavior can lead to delayed wound closure or exaggerated fibrosis [5]. Conditions associated with altered fibroblast behavior include chronic ulcers in metabolic diseases, chronic infections, and hypertrophic scarring, particularly in the context of high mechanical force [11,12]. Mechanoreceptors transduce tension across wounds, influencing fibroblast migration, orientation, elongation, and proliferation [2,13,14,15,16]. Additionally, mechanical loading reduces myofibroblast apoptosis, contributing to hypertrophic scar formation [11].

In response to the demonstrated relationship between mechanical force and fibroblast behavior, various synthetic external tension off-loading devices and dressings have been developed for wound healing and scar management [11]. As cellular pathways associated with mechanotransduction have been identified, new avenues have been opened for targets of pharmaceutical drug development. Future therapies may target sites of mechanotransduction, like mechanosensitive calcium-dependent channels, Rho-GTPases, and the integrin-like kinase-phosphoinositide 3-kinase/protein kinase B pathway, which are activated by stretch and integrin stress [17,18,19]. Similarly, focal adhesion kinase (FAK), a non-receptor tyrosine kinase, is phosphorylated upon mechanical injury, initiating inflammatory signaling, fibroblast recruitment, and collagen production [20]. Downstream pathways, such as those associated with wnt, β-catenin, and yes-associated protein (YAP), alter transcription and initiate profibrotic processes [21]. There is significant heterogeneity in dermal fibroblast responses to wnt/β-catenin signaling pathway, which has previously been associated with the role of wound macrophages on hair follicle stem cell activation [22,23,24,25,26,27,28,29].

In parallel, natural compounds from diverse biological reservoirs have been investigated for their capacity to promote wound tissue regeneration and repair [30,31,32,33]. These natural compounds can be used directly as treatments or as drug carriers for the delivery of other therapeutics. Natural bioactive agents have recently received attention in wound management due to their efficacy and minimal side effects, though the history of natural compounds in the field of wound healing is rich and diverse. Although natural remedies have been used for thousands of years, recent advancements in skin delivery, the understanding of phytochemistry and biologic activity, and the characterization of the mechanism of action through modern scientific techniques, such as high-throughput transcriptomic and proteomic analyses, 3D organotypic cell culture models, live cell imaging, and integrative omics approaches, have allowed for a deeper appreciation of how their therapeutic effects are often mediated by fibroblast behavior [34,35,36,37,38,39]. For example, honey’s phenolic compound constituents have been shown to modify fibroblast behavior, enhancing wound healing outcomes [31,40]. Understanding how these compounds affect wound repair and fibroblast modulation is considered crucial [41]. This review aims to highlight natural compounds capable of modulating fibroblast behavior, consequently facilitating wound regeneration and repair. Additionally, it delves into the realm of biomimetic engineering, where researchers draw inspiration from nature to design materials and devices that replicate physiological cues essential for optimal wound repair (Figure 2).

## 2. Natural Compounds

### 2.1. Botanicals and Plant-Derived Products

The utilization of botanicals in the context of wound healing has a rich legacy within medicinal practices, with their therapeutic properties well documented in the ethnobotanical literature [31,42,43,44,45,46]. This reliance on natural remedies persists today, with an estimated 70–80% of the global population depending on plant-derived medicines for disease treatment [47]. Moreover, the prevalence of non-healing wounds, affecting approximately six million individuals worldwide, presents a significant healthcare challenge and an economic cost of USD 3 billion annually [45]. Given the costs and unfavorable side effects associated with conventional medicines, the quest for natural compounds with healing properties is a prominent focus of green chemistry [45,47].

Notable advantages of these natural remedies include their cost-effectiveness, wide availability, and reduced risk of adverse side effects [35]. Consequently, there has been a growing interest in understanding the intricate mechanisms underlying the therapeutic actions of herbal medicines in tissue regeneration and repair processes, with many attributing the reparative and regenerative properties of medicinal plants to their bioactive phytochemical constituents influencing mechanotransduction processes (Table 1. Natural compounds) [31,35,45,48]. Premarathna et al. tested 23 seaweed species that have historically been used medicinally, almost all of which contained bioactive compounds such as saponins, tannins, flavonoids, and alkaloids. In vitro cell migration assays found the *Sargassum illicifolium* and *Haliimeda opuntin* extracts to be the most efficacious, a result that was mirrored by expedited healing during the in vivo treatment of excisional wounds. This result was connected to the ability of these compounds to enhance fibroblast proliferation and increase the action of myofibroblasts [49].

Several botanicals have been recognized as potential regulators of fibroblast behavior and mechanotransduction pathways related specifically to TGF-β1. Sarıçiçek (*Achillea biebersteinii Afan*), a member of the Asteraceae family high in phenolic acids and flavonoids, widely utilized for wound healing in Mediterranean regions, inhibits scar formation by modulating the expression of TGF-β1 and basic fibroblast growth factor (bFGF) at both the gene and protein levels in murine embryonic fibroblasts [45,50]. These growth factors promote wound healing via fibroblast proliferation and granulation tissue formation [10]. Similarly, traditional Chinese herbs rich in saponins and flavonoids, such as *Astragali Radix*, *Rehmanniae Radix*, and *Centella Asiatica*, have been shown to modulate human skin fibroblast migration activities through the TGF-β1 pathway, ECM synthesis through the Suppressor of Mothers Against Decaptentaplegic (Smad) pathway, and cell motility through the Ras/MAPK (non-Smad) pathway [52,70].

Flavonoids such as quercetin and polysaccharides such as APS2-1 from *Astragalus membranaceus* upregulate the expression of TGF-β1, bFGF, and epidermal growth factor (EGF) [31,67]. Additionally, terpenoids like terpinolene and alpha phellandrene, along with alkaloids like taspine, are associated with enhanced fibroblast migration. Terpinolene and alpha phellandrene as well as curcuminoids and polyphenols such as tannins have demonstrated the capacity to enhance fibroblast proliferation, while taspine plays a crucial role in promoting the expression of keratinocyte growth factor (KGF), a member of the FGF family [31,58,63,64,65]. Furthermore, flavonoids including naringin, morin, and icariin have been shown to accelerate collagen deposition [46,86]. These properties of botanical plants and their derivatives highlight their potential significance as mechanotransduction modulators capable of augmenting tissue regeneration and repair.

### 2.2. Bacterial-Derived Products: Botulinum Toxin A

Established clinical applications of botulinum toxin are diverse, including wrinkle improvement, facial and body contouring, as well as treatment for migraines, hyperhidrosis, and cervical dystonia [87]. Derived from the anaerobic gram-positive bacteria Clostridium Botulinum, Botulinum toxin exhibits seven serotypes, with type A (BTA) being the most widely employed clinically [88]. BTA functions by blocking the presynaptic release of acetylcholine, leading to neuromuscular junction inhibition and temporary muscle paralysis lasting from two to six months. Recently, the compound’s paralytic capabilities have been aimed at alleviating tension during wound healing, consequently minimizing scar formation.

In the context of scars and wound healing, BTA, through its mechanism of reducing tension on the dermis caused by underlying musculature movement, shows promise in enhancing healing outcomes. Numerous studies support the efficacy of BTA injections in enhancing the aesthetic and functional aspects of surgical scars [89,90]. Gassner et al. observed substantial cosmetic improvements in facial excision scars following BTA treatment in both primates and humans [90], while Yue et al. demonstrated significant reductions in wound tension, improved functional outcome scores, and enhanced facial scar appearance with BTA injection [89]. Besides its mechanical impact, various studies have highlighted a direct or indirect influence of BTA on fibroblast proliferation and differentiation [89,91].

In vitro studies indicate that BTA may reduce TGF-β activity by inhibiting the expression of TGF-β1, as well as lowering the transcription and expression of other profibrotic cytokines like connective tissue growth factor (CTGF). TGF-β, associated with hypertrophic scars, stimulates fibroblasts, increases collagen deposition, and inhibits fibroblast apoptosis [92]. Consequently, BTA may inhibit scar hyperplasia and treat hypertrophic scars by reducing fibroblast proliferation [93,94]. Liu et al.’s study demonstrated a dose-dependent cellular reduction in wounds [92]. Zhou also revealed that BTA could reduce angiogenesis by inhibiting the expression of related cytokines, such as VEGF, in hypertrophic scars [95]. Additionally, BTA modulates the differentiation of fibroblasts into myofibroblasts, as seen in in vitro studies indicating a decrease in alpha-smooth muscle actin mRNA and protein levels in fibroblasts, resulting in reduced myofibroblast differentiation and wound contractility [92]. Therefore, beyond its mechanical off-loading properties, BTA is a valuable natural compound in wound healing and scar management due to its lesser-known effects on fibroblast behavior.

### 2.3. Animal-Derived Products

The therapeutic potential of compounds derived from insects and aquatic animals in wound healing are being explored more in depth [51,74]. Insects, historically used in folk medicine, provide royal jelly, beeswax, pollen, and propolis, which have been used to treat diabetes and arthritis [51,71]. Recent studies have shed light on the remarkable ability of these insect-derived medicinal products, such as honey, maggot excretions/secretions (ES) from *Phaenicia sericata*, and sericin from Bombyx mori silkworm cocoons, to modulate tissue regeneration and repair through mechanotransduction pathways [51,71,72,73]. Notably, honey and maggot ES have been found to stimulate fibroblast proliferation, with honey additionally promoting fibroblast migration [51,71,73]. On the other hand, sericin has demonstrated its efficacy in preventing scar tissue formation by decreasing the expression of TGF-β1 and TGF-β3 [96]. While the benefits of these insect-derived products are well established in the history of wound healing, recent efforts have focused on incorporating them in novel ways, such as in the development of natural rubber latex membranes infused with three types of propolis [97].

The marine ecosystem has also emerged as a valuable source of biologically active compounds conducive to tissue regeneration and repair [51,74]. Marine collagen and sea cucumbers exhibit the capacity to invoke keratinocyte and fibroblast migration, improve skin vascularity, and modulate mechanotransduction pathways [33,51,72,74]. Both marine collagen and sea cucumbers have shown the ability to promote fibroblast chemotaxis, while sea cucumbers can contribute to the breakdown of ECM proteins and indirectly induce fibroblast proliferation through the activation of T-lymphocytes, which secrete TGF-β [33,51,72,74]. Consequently, these animal-derived products have been recognized as significant regulators of mechanotransduction and fibroblast behavior.

### 2.4. Exogenous Growth Factors and Growth Factor-Rich Products

The wound healing process is orchestrated through the coordinated interplay of various cells, guided by a complex signaling network of growth factors (GF) [76]. Recent years have witnessed a growing body of evidence supporting the use of exogenous GFs and GF-rich products to enhance wound healing outcomes [72,77,98,99].

These studies have elucidated specific GFs that play critical roles in modulating mechanotransduction and fibroblast behavior in pathways crucial for tissue regeneration and repair. Specifically, TGF-β, FGF, and PDGF have emerged as key drivers of phenotypic changes in fibroblasts, leading to their transformation into myofibroblasts and promoting wound contraction [75,76]. TGF-β1 and TGF-β2 can stimulate the production and deposition of ECM, while bFGF inhibits TGF-β1/Smad-dependent pathways [77,79]. This regulation of ECM synthesis and degradation is achieved through collagen distribution, α-smooth muscle actin (α-SMA) expression, and TGF-β1 modulation [77,79]. TGF-β3, PDGF, insulin-like growth factor (IGF)-1, EGF, and neuregulin 1 (NRG1), an EGF family member, have all been validated as inducers of fibroblast proliferation [63,76,77,80,81]. PDGF, EGF, and NRG1 exhibit the ability to induce fibroblast chemotaxis to wound sites, while both PDGF and IGF-1 stimulate collagen synthesis, further enhancing the wound healing process [77,78,80,81].

In addition to these GFs, various GF-rich products, such as Platelet-Rich Plasma (PRP), Platelet-Rich Fibrin Matrix (PRFM), and Decellularized and Dehydrated Human Amniotic Membrane (DDHAM), have demonstrated potential in modulating mechanotransduction during wound healing due to their GF constituents [77,82,83,84,85]. PRP is rich in PDGF and TGF-β1, and both PRFM and DDHAM products stimulate the secretion of PDGF, TGF-β1, EGF, and FGF-2 [77,82,83,84]. These findings underscore the diverse array of exogenous GFs and GF-rich products that hold promise in influencing mechanotransduction pathways integral to enhancing tissue regeneration and repair outcomes.

## 3. Biomimetic Engineering of Biomaterials

Biomimetic engineering involves the development of technologies inspired by natural structures and biological phenomena. These methods are applied to numerous fields including art, architecture, engineering, and medicine. Well-known examples in the industrial world include burr-inspired Velcro, bird-inspired airplanes, and sharkskin-inspired Olympic swimsuits [100]. In the medical field, drug-delivering nanoparticles (NPs) mimic vesicles and liposomes [100,101]. Additionally, biomimetic engineering plays a promising role in tissue engineering, where scaffolds mimic tissue structures and/or cell–cell interactions. Biomimetic applications have been demonstrated to improve bone, cardiac, nerve, hair, and skin tissue regeneration, among others [102,103,104,105].

Recent advancements in bioengineering have focused on the development of biomimetic dressings that closely resemble the natural skin environment, aiming to enhance wound healing [106,107]. Conventional dressings often require frequent re-application, potentially hindering re-epithelialization and prolonging inflammation [107]. Biomimetic dressings, designed to mimic the ECM of healthy skin, seek to replicate the skin’s inherent regenerative mechanisms [108,109,110,111,112,113]. The ECM plays a crucial role in influencing fibroblast proliferation, differentiation, migration, and adhesion [113,114,115,116,117]. While biomimetic dressings also exhibit the potential for anti-microbial activity, drug delivery, and temperature/humidity regulation [107,118], this review focuses on how they specifically regulate fibroblast activity to reduce fibrosis by reproducing cell–ECM interactions in healing wounds [113,114,119].

Electrospun nanofibers are the preferred method to mimic the natural wound microenvironment. However, other technologies discussed elsewhere in this review, such as hydrogels and NPs, similarly possess favorable biomimetic properties. Electrospun nanofibers exhibit a high surface area and modifiable topography, which make them favorable biomimetic substrates [115]. Electrospun nanofiber dressings can be composed of many different biocompatible synthetic or natural polymers. Indeed, over 100 distinct polymers have been successfully electrospun. These polymers must fulfill several criteria for successful electrospinning, including a high molecular weight, the ability to be dissolved in a low-dielectric-constant solvent, and optimal electrical conductivity [115]. In general, natural polymers such as chitosan, silk fibroin (SF), and collagen are selected because of their bioactivity contributing to a favorable wound microenvironment. However, natural polymers must typically be mixed with synthetic polymers such as poly(lactic acid), polyurethane (PU), or poly-caprolactone (PCL) to achieve the mechanical parameters necessary for electrospinning [115]. Each polymer exhibits unique properties, allowing the electrospun nanofiber meshes to be tailored to the goal of the scaffold. For example, human skin fibroblasts bind favorably to PU, poly(3-hydroxybuturate-co-3-hydroxyvalerate) (PHBV) has high elasticity and inhibits myofibroblast differentiation, PCL has high plasticity and ductility but lower cell adhesion rates, and SF is highly biocompatibility with low immunogenicity [119,120,121]. Variations in nanofiber polymers, as well as variations in mesh alignment, pore size, and fiber diameter impact fibroblast activity and scar formation. Wounds treated with nanofibers have been found to have a higher density of fibroblasts, as well as inflammatory and epithelial cells, and vascular tissue [122].

Perpendicularly aligned electrospun nanofibers offer topographical cues, relieving wound tension and preventing myofibroblast differentiation and scar formation [113]. In healing wounds, collagen fibers align linearly with wound tension. Both mechanical tension and topographical cues promote myofibroblast differentiation and proliferation, leading to pathological scar formation [114,117,123,124]. A dressing developed by Chen et al. using PCL and SF aligned perpendicularly disrupted pathologic topographical cues, resulting in minimal scar formation. Compared to parallel cues, perpendicular topographical cues reduced fibrosis-related proteins such as FN, TGFβ-1, and α-SMA, and the mechanosensitive protein YAP-1. This reduction indicates a decrease in the differentiation of resting fibroblasts into pro-fibrotic myofibroblasts [113].

Kim et al. demonstrated how a PHBV biodegradable electrospun nanofiber mesh, resembling the fibrous matrix of skin ECM, reduced mechanical stress on wounds. The mesh’s pores allowed cell migration, inducing fibroblast proliferation while reducing myofibroblast differentiation. This resulted in the ordered deposition of a new ECM and reduced scar formation in healed wounds [119].

Modifying the diameter of electrospun fibers alters the topography in healing wounds, influencing fibroblast migration and proliferation [125]. Gao et al. observed increased cell migration and expression of wound-healing genes on larger-diameter PU/Poly(lactide-co-caprolactone)/Polyethylene oxide nanofibers, while smaller diameters increased fibroblast proliferation. Although limited to in vitro experiments, these findings highlight the importance of topographical cues on fibroblast activity, offering an additional tunable variable in biomimetic dressing design.

Electrospun nanofibers, mimicking skin ECM cues, influence fibroblast proliferation, migration, and differentiation. Fibroblasts are crucial in early wound healing, but their dysregulation contributes to fibrosis [114]. Ideal biomimetic dressings express ECM-like cues that enhance fibroblast migration and proliferation while limiting myofibroblast differentiation, leading to faster healing with limited scarring. These studies showcase how tunable characteristics of biomimetic dressings, such as polymer type, fiber alignment, and fiber diameter, influence fibroblast phenotypes and fibrosis development in healing wounds.

## 4. Natural Compound Delivery Systems

While preclinical in vivo findings regarding the use of natural compounds to modulate fibroblast behavior are often promising, numerous compounds exhibit insufficient efficacy regarding wound healing in clinical trials. This discrepancy may stem from various issues, such as poor bioavailability within the wound environment, challenges in drug retention, and difficulties in penetrating the stratum corneum, among other obstacles related to drug delivery challenges in vivo [126]. Indeed, the journey of topical compounds to penetrate the wound site is limited by several physiological barriers, and passive drug diffusion into the skin is generally impeded by the outermost lipophilic layer of the skin, the stratum corneum. This tight layer is composed of lipids including triglycerides, cholesterol, and free fatty acids that are linked by covalent bonds with corneocytes and provide a defensive barrier to the external environment [127].

Furthermore, only moderately lipophilic compounds can perform transcutaneous passage [128]. To address these limitations, both natural and synthetic biomaterials have been developed to fuse with stratum corneum, disrupt the skin surface, or accumulate in the hair follicles to act as a long-term reservoir. These biomaterials must be delivered in a controlled manner to maintain therapeutic drug concentrations in order to target specific cells and cell components (e.g., fibroblasts) to achieve a robust local effect while minimizing potential side effects, (Table 2. Delivery systems). However, the enhancement of transdermal drug delivery could be achieved using penetration enhancers, skin disruption, or physical techniques. Innovative formulation designs using new technologies have emerged as effective, reliable methods for the delivery of natural compounds [129,130].

The incorporation of biomaterial-based scaffolds into the field of wound healing has gained prominence since the late 1990s, coinciding with the initial definition of “tissue engineering” [159]. These scaffolds, constructed from various synthetic and natural polymers, possess key features that are ideal for the wound healing process. Notably, they can be functionalized with various agents to enhance wound healing and serve as effective delivery systems on the macroscale (Figure 3) [160]. Rapidly, new therapeutic alternatives using engineering have been developed at the micro and nanoscale. In addition to the size difference, there is a range of physical and chemical properties that differ between those particles, such as melting point, dissolution, and solubility, allowing them to be used in different indications.

These drug delivery systems not only act as physical support or barriers for wounds but also actively release therapeutic compounds, improving bioavailability, optimizing pharmacokinetics, and reducing dosing frequency, thus enhancing and optimizing the healing process [161].

### 4.1. Micro Delivery Systems

Micro delivery systems are designed to overcome limitations relative to the ability of larger particles to penetrate deeply into tissues for maximum effectiveness at the cellular level [162]. However, these microscale systems can transport relatively large quantities of natural compounds due to their larger relative size and can be produced at a lower cost compared to nano-delivery systems. 

Adhirajan et al. developed a gelatin microsphere conjugated to an MMP inhibitor and loaded with doxycycline into a collagen dressing. Their ability to reduce the MMP level was assessed in a rat wound model [163]. Other authors produced chitosan microparticles loaded with EGF and VEGF and suspended them in dextran hydrogel for application to burn wounds. This new therapy provided a beneficial effect in a burn wound model in rats; however, repeated applications were needed, indicating a potentially reduced duration effect [164].

### 4.2. Nano-Drug Delivery Systems

Following the advent of nanotechnology and recognizing the prevalence of biological phenomena at the nanoscale, numerous nano-drug delivery systems have been introduced to the realm of skin and wound regeneration [165,166].

Due to their microscopic size (1–100 nm), these biomaterials can exert unique effects at the molecular and cellular levels, with some demonstrating the ability to enter the cytoplasmic space or activate specific transport mechanisms [166]. Nano-drug delivery systems (NDDS) exhibit biocompatibility, biodegradability, non-toxicity, non-immunogenicity, and the ability to create a moist environment beneficial in the field of wound healing. Additionally, they possess inherent anti-bacterial properties, a high surface area-to-volume ratio, and can encapsulate both hydrophilic and hydrophobic drugs [166]. NDDS can integrate natural bioactive molecules, shielding them from degradation, delivering them to the application area, and sustaining their release. This capability enhances the therapeutic efficacy of drugs, reduces the frequency of administration, lowers overall costs, and improves patient compliance [167,168]. At present, nanomaterials stand out as the most promising delivery systems due to their distinct advantages, offering alterations in physical and chemical properties in the field of wound healing.

Bioactive molecules incorporated into NDDS are diverse, encompassing anti-microbial agents, growth factors, and genes, and can be either of natural or synthetic origin. Various macroscopic nanomaterials, including NPs but at a macroscopic size (e.g., electrospun nanofibers, nanosheets, nanoemulsions, carbon nanotubes-based, or graphene-based nanocomposites), and nano-sized biomaterials (e.g., NPs, ions, molecules, nucleic acids, functional peptides, proteins, oligosaccharides, or polysaccharides), have been documented [169]. As a result, natural compound-eluting nanomaterials or natural encapsulated nanobioactive molecules represent a particularly promising avenue for synergistically enhancing the healing process of wounds by acting on fibroblasts at the nano-scale [161].

### 4.3. Liposomes and Transfersomes

Liposomes, characterized as bilayer amphiphilic vesicles, have found extensive use in wound dressings and are emerging as promising nanocarriers [170]. In addition to safeguarding hydrophilic drugs like growth factors, hormones, and metabolites, they create a conducive moist environment for wound healing [134,167]. The topical formulation of insulin in the field of wound healing has encountered multiple problems, as it cannot be delivered safely and in a controlled manner. To overcome this limitation, Dawoud et al. devised insulin-loaded chitosan nanoparticle liposomes, effectively prolonging insulin release and demonstrating improved wound healing in human patients [135]. In another study, Xu et al. developed a unique liposome with an SF hydrogel core incorporating bFGF, enhancing the stability of bFGF and expediting wound healing by stimulating angiogenesis [171]. Additionally, usnic acid, a type of lichen metabolite first isolated in 1844, has been utilized in a liposome-embedded gelatin-based membrane by Nunes et al. and applied in a porcine burn wound model. This product was compared to two different standard dressings for burn wounds. The authors observed a highly fibroblastic component in the granulation tissue of the treated group on day 18, along with extensive condensation of thick collagen fibers and narrower interfibrillar spaces compared to those of the other groups, indicating induced fibroblast proliferation and synthesis [133].

Transfersomes, deformable liposomes featuring an edge activator such as sodium, have been designed to enhance their ability to traverse the stratum corneum and reach deeper layers of the epidermis [167]. Manconi et al. innovatively developed a self-assembling core–shell gellan transfersome, functioning as a nanohydrogel, to encapsulate baicalin for wound treatment in mice. Their findings demonstrated that these delivery systems could optimize skin drug deposition, with approximately 11% of baicalin retained in the entire skin, including 8% in the dermis, ultimately promoting enhanced skin regeneration. Baicalin-loaded formulations provided the highest Tumor Necrosis Factor alpha (TNF-α) and TGF-B1 inhibition in a mice wound model, which led to significant improvements in wound healing [136].

### 4.4. Lipid Nanoparticles

Introduced to overcome the limitation of liposomes, lipid NPs are also receiving increasing interest because they permit the administration of different drugs, such as growth factors or anti-bacterial peptides [145]. Lipid NPs are prepared with physiological lipids or lipid molecules without any organic solvent and can be either solid lipid NPs or nanostructured lipid carriers [167]. Gainza et al. produced solid lipid NPs and nanostructured lipids loaded with rhEGF, and tested them in vitro and in vivo in a murine full-thickness wound model, which resulted in expedited wound closure and re-epithelialization, primarily through increased fibroblast proliferation [167].

### 4.5. Polymeric Nanoparticles

Polymeric NPs serve as effective carriers for encapsulating natural agents, providing controlled release and protection from degradation by proteases in the wound microenvironment [138,157]. Poly(lactic-co-glycolic acid) (PLGA) has demonstrated its potential in promoting wound healing by expediting neovascularization. Recent developments include polymeric NPs incorporating anti-microbial agents. Chereddy et al. introduced a PLGA nanoparticle loaded with the defense peptide LL37, showcasing its modulation of wound healing, angiogenesis, and infection prevention. In the context of full-thickness wounds, the application of PLGA-LL37 NPs significantly accelerated the healing process by influencing collagen deposition, epithelialization, and neovascularization. By day 10, the extent of collagen deposition was higher in the treated group, and the deposited collagen was thicker and denser [156].

Manukumar et al. conducted a study on thymol-loaded chitosan silver NPs (AgNPs) against biofilm-associated proteins in methicillin-resistant *Staphylococcus aureus* (*S. aureus*), demonstrating excellent anti-bacterial activity and anti-biofilm properties without inducing toxicity. These findings underscore the potential application of this drug in the realm of wound healing [139].

### 4.6. Inorganic Nanoparticles

Inorganic natural NPs, encompassing metal NPs, mineral NPs, biogenic NPs, ceramic NPs, and more, are prevalent in various environments such as geological formations, soils, and biological systems [140,172]. Natural NPs exhibit an improved duration of action and biocompatibility compared with other nanoparticles or pure drugs [158,173,174]. Their utility extends to the field of wound healing, where they serve as intrinsic anti-bacterial agents, wound healing enhancers, and drug delivery systems [167]. AgNPs, a widely utilized inorganic variant, play a crucial role in preventing infections and significantly enhancing the healing of chronic wounds by disrupting the intracellular activity and cell membrane of bacteria [170].

To enhance full-thickness dermal wounds in mice, Chigurupati et al. developed water-soluble cerium oxide NPs (metal NPs). In vitro, the migration and proliferation of fibroblasts was increased compared with those in the control group, and the growth rates of both fibroblasts and keratinocytes were significantly increased in wounds treated by NPs. In vivo, the rate of wound closure was significantly higher in mice treated with NPs, and immunostaining showed a higher rate of α-SMA expression and thus an enhancement of myofibroblast differentiation in those wounds [141].

Ziv Polat et al. utilized maghemite NPs to stabilize thrombin, applied clinically for topical hemostasis and wound healing, in an incisional wound model [140]. Results indicated that the NP-treated group exhibited the fewest inflammatory cells, the least granulation tissue along the surgical scar, and the highest values of skin tensile strength compared with the free thrombin group. The evidence supported the notion that thrombin-bonded maghemite NPs significantly advanced the wound healing stage and achieved better overall healing quality [140].

Kim et al. used gold NPs coated on a hydrocolloid membrane (HCM) to treat wound injuries in a rat model. The authors showed a decreased time to wound closure as well as an increased expression of collagen and a decreased expression of collagen-degrading enzyme at the early stage of the wound skin repair process [143].

### 4.7. Nanofibrous Structures (Nanofibers/Nanoscaffold)

Nanofibers, composed of both natural and synthetic continuous polymer chains, serve as versatile materials, forming nanofibrous sheets or 3D scaffolds applied in tissue engineering [157]. These nanofibrous structures, designed to emulate the ECM, create conducive conditions for cell attachment and interactions with therapeutic agents. The nanofibrous structure facilitates the transfer of various therapeutic agents, including GFs, nucleic acids, and anti-microbial agents [157]. Electrospinning stands out as the most widely adopted technique for nanofiber production.

To address the poor solubility of andrographolide in aqueous media, Jia et al. incorporated mesoporous silica NPs loaded with andrographolide into a PLGA matrix. Unlike lipid NPs previously used for andrographolide encapsulation and incorporation into sponge scaffolds for topical applications, the use of silica NPs was preferred due to their high drug encapsulation capacity, and PLGA was chosen for its good biocompatibility, adjustable mechanical properties, and tunable degradation rate. The PLGA/Andro-MSNs demonstrated an accelerated wound healing rate with increased epidermal cell adhesion and a reduction in the inflammation process compared with other samples, showcasing the effects of sustained andrographolide release [146]. The incorporation of mesoporous silica NPs into PLGA also enhanced the hydrophilicity of the nanofibrous membranes [146].

Shan et al. evaluated the effectiveness of an SF/gelatin nanofibrous dressing loaded with astragaloside IV on deep partial-thickness burn wounds [147]. This dressing not only promoted cell adhesion and proliferation with good biocompatibility in vitro but also demonstrated efficacy in wound healing and reduced scar formation in vivo in mice [147].

### 4.8. Nanohydrogel and Hydrogels Loaded with Nanoparticles

While hydrogels excel at effectively encapsulating, protecting, and sustaining the release of compounds, the integration of NPs into hydrogels has emerged, leveraging their benefits synergistically within a single system [175]. To enhance hydrogel adhesion to the skin and investigate the impact of phenytoin on wound healing, Cardoso et al. designed chitosan hydrogels loaded with nanoencapsulated phenytoin. In a porcine wound model treated with phenytoin nanocarriers, the authors observed a significant presence of fibroblasts and collagen deposition on day 6, indicating the proliferation and synthetic activity of fibroblasts [153].

Mahmoud et al. characterized and loaded gold NPs into thermosensitive hydrogels. Demonstrating the advantageous combination of the properties of gold NPs and hydrogel stability, the authors showcased excellent prolonged drug release behavior. In a murine wound model, the treated group exhibited enhanced skin re-epithelialization, granulation tissue, vascularization, and collagen deposition. Furthermore, in vitro studies highlighted the drug’s exceptional anti-bacterial activity and its impact on the expression of inflammatory and anti-inflammatory mediators [154].

Aly et al. formulated hydrogel loaded with polymeric NPs of simvastatin for topical application. Their study revealed that 81% of simvastatin was released into the skin after 24 h. Histopathological assessments confirmed a notable wound healing effect, characterized by the formation of a normal epithelial layer by day 11 [157].

Organic NPs have also been designed for use in wound healing. Shalaby et al. transformed collagen derived from tilapia fish scales and incorporated it into PRP gel, resulting in a novel nanomaterial. This demonstrated faster re-epithelialization and wound closure in vivo. Treatment with collagen NPs improved wound contraction and increased the number of myofibroblasts in the wound environment [158].

Nanohydrogels are often engineered to possess enhanced properties, including increased surface area, enabling them to encapsulate a wide variety of drugs with high compatibility and efficacy. This significantly affects skin regeneration due to their improved interaction with biological tissues and cells, including fibroblasts, at the nanoscale.

Dehkordi et al. utilized nanocrystalline cellulose human leukocyte antigens enriched with granulocyte-macrophage colony-stimulating factor (GM-CSF)-loaded chitosan to control the delivery of this growth factor to the wound site in rats. By day 13 of wound healing, the treated group exhibited significantly higher granulation tissue formation, attributed to fibroblast proliferation and differentiation. Treated wounds also displayed a lower inflammatory reaction, enhanced epithelialization, and a reduced wound closure time [150].

## 5. Future Directions for the Field

As natural compounds are introduced to the field of wound healing, future research should prioritize innovative and effective therapeutic strategies that exploit recently described signaling pathways, gene expression patterns, and interactions crucial in influencing fibroblast behavior. The current therapeutic landscape of biomaterials in wound healing, including many of the biomaterials previously discussed, often consists of a single therapeutic and lacks the dynamic, multi-component delivery necessary for effective wound healing. A new area of exploration lies in scaffolds that can respond to the dynamic wound microenvironment to optimize therapeutic outcomes [176].

### 5.1. Micro-Environment-Responsive Biomaterials

Responsive biomaterials designed for micro-environments exhibit adaptability to various endogenous biological/biochemical (i.e., reactive oxygen species (ROS), pH, enzymes, glucose, and glutathione), and exogenous physical conditions (temperature, mechanical forces, and pressure), or a combination mode to optimize therapeutic responses [176]. For instance, ROS-cleavable linkers, ROS-responsive polymers, and ROS-sensitive indicators can be used for the real-time monitoring of ROS and modulating redox balance in the wound environment [176,177]. Wound pH can vary depending on the stage of healing, infection status, and the presence of exudates. Under the influence of hypoxia and lactic acid stimulation, acute wounds are induced to become acidic [176]. In contrast, chronic wounds consistently maintain an alkaline pH within the range of 7.15–8.90 [178]. Biomaterials responding to pH could release drugs or bioactive agents as appropriate according to the characteristics of the wound [179,180]. Additionally enzymatic stimuli, such as MMPs, are involved in various aspects of wound healing, including ECM remodeling, cell migration, and angiogenesis [181]. Chronic wounds, particularly diabetic ulcers, are characterized by elevated glucose levels. Glucose-responsive biomaterials can be designed to release insulin or other therapeutic agents in response to high glucose concentrations [182]. However, endogenous physiological stimuli exhibit a semi-passive nature, creating a challenge for controlled dosage regulation.

Conversely, drug delivery dressings activated by exogenous stimuli can be modulated through mechanical force, temperature, light, ultrasound, magnetic, and electric fields, providing more predicted responses. Li et al. developed a thermo-sensitive mechanically active hydrogel that actively contracted the tissue surrounding the wound in response to body temperature in a rat model [183]. Histological analyses revealed that the hydrogel treatment mechanically contracted the wound, reduced inflammation, and improved wound healing quality. 

Pressure-sensitive hydrogels with anti-bacterial properties can detect skin pressure and transmit electrical signals, making them ideal for monitoring and treating pressure ulcers [184]. Another innovative wound dressing material is based on moisture-adaptive contractile fibers. Dong et al. utilized moisture-adaptive contractile fibers fabricated through a simple wet spinning technique and knitted into textiles to create wound dressings that conform to the shape of the wound [185]. These fibers absorb wound tissue fluid, elongate to create tensile force, and, upon drying, contract to promote wound closure, accelerating healing and preventing infection [185].

### 5.2. Smart Dressings

While existing biomaterial systems typically monitor only one parameter at a time, future research aims to develop biomaterials equipped with multi-sensor capabilities capable of adapting and responding to various biomarkers within the dynamic wound microenvironment. This involves integrating real-time sensors into the wound microenvironment to enable the precise and controlled delivery of therapeutics, referred to as smart dressings. For example, Zhang et al. utilized an innovative integrated smart dressing consisting of three layers: a biomimetic nanofiber membrane, a gelatin methacryloyl and β-cyclodextrin ultraviolet crosslinked hydrogel, and a sensor chip [186]. The nanofiber membrane promotes cell migration and skin regeneration, while the hydrogel facilitates the controlled release of growth factors and cytokines for tissue remodeling, granulation tissue remodeling, and angiogenesis [186]. The sensor ensures accurate, stable measurements, and biocompatibility for optimal dose release [186]. Liang et al. demonstrated the use of pH/glucose dual-responsive metformin-released hydrogel dressings in a rat diabetic model [187]. These dressings respond to the physiological conditions of diabetic wounds, releasing metformin in both acidic and high-glucose environments. The hydrogel displayed superior self-healing, anti-oxidant, anti-inflammatory, and pro-angiogenic properties, effectively promoting the healing of chronic diabetic foot wounds in rats [187]. 

Smart dressings have immersive potential for clinical applications in telemedicine, artificial intelligence-based diagnosis, and personalized precision medicine for wound healing. However, the early stages of wound infection often exhibit subtle changes in biomarker levels that are susceptible to external noise, guiding inaccurate sensor responses. Therefore, future biomaterials and wearable smart dressings will likely work to optimize accuracy, sensitivity, and stability. 

### 5.3. Stem Cells

While bone marrow mesenchymal stem cells have been extensively studied, there is a growing interest in exploring the potential of stem cells from fat tissue and hair follicles for regenerating wounds [188,189,190]. Natural compounds and biomaterials can serve as conductors to guide stem cell differentiation and functionality. The combined effects of stem cells and fibroblasts in improving skin healing and reducing scarring, especially with different cell lineages of mesenchymal stem cells (MSC), have been extensively explored. Vojtassák et al., 2006, conducted a study on chronic diabetic foot ulcers, combining autologous fibroblasts and MSCs on a biodegradable collagen membrane, resulting in decreased wound size, increased vascularity, and dermal thickness [191]. Yates et al. produced a similar result by experimenting on a full-thickness excisional wound model in mice to assess the effects of simultaneously applying MSC and fibroblasts in a collagen-tenascin-C matrix on wound healing and scarring [192]. The results confirmed that MSC and fibroblast co-transplantation increased the expression of collagen, fibronectin, and other matrix-related genes, and decreased the expression of pro-inflammatory cytokine. Moreover, MSCs can influence collagen organization, leading to a thinner, more flexible scar with a basket weave structure compared with the dense, disorganized scar tissue typically formed in control wounds. This improved structure is associated with greater tensile strength in the healed tissue [193]. Nonetheless, several challenges confront current cell-based wound therapies, including issues like immune rejection, complex treatment protocols, an incomplete understanding of the underlying molecular pathways, and the intricacies of managing cell viability and function.

## 6. Limitations

The integration of traditional healing agents with modern natural or synthetic biomaterials, such as nanofibers containing silver nanoparticles, aloe vera-loaded alginate hydrogels, propolis in dressing films, and hydrogel sheets containing honey, has been explored in various studies [194,195]. Although they have demonstrated efficacy in clinical trials, testing primarily involved the direct application of agents as topical solutions or in combination with gels and dressings [194], with a limited exploration of their effectiveness in conjunction with more modern drug delivery systems such as nanoparticles, biomaterials, and smart dressings.

As previously noted, nanofiber-based substrates have demonstrated the ability to enhance wound healing by targeting pathways influenced by fibroblasts. However, the efficacy of most nanofibrous materials in wound healing has primarily been studied in 2D cultures, which lack the complexities of the wound microenvironment observed in vivo 3/12/2024 9:24:00 AM. Since biologically significant signaling pathways, particularly those involving the interplay between adhesion and growth, function optimally when cells are organized spatially within 3D tissues, there has been a growing trend in current and future studies towards transitioning from 2D cell culture to 3D bioscaffolding [196,197]. The risk of over-proliferation and scar formation has already begun to be addressed by embedding nanofiber substrates with drugs [198] or cell components [199] that inhibit fibroblast differentiation into myofibroblasts. For instance, electrospun nanofibrous scaffolds, crafted from a combination of poly(ε-caprolactone), gelatin, and palmatine, were tested on wounds in rabbit ears and demonstrated promising anti-scarring properties beyond their influence on fibroblasts and collagen density [198]. Scar elevation index measurements and histological analyses revealed a significant reduction in hypertrophic scaring formation compared with that in the control group, and a shorter wound healing duration [198]. Nevertheless, there are still currently limited biomaterials investigated for anti-scar and skin regeneration benefits in clinical practice. 

Furthermore, many experiments conducted on biomaterial applications in wound healing have utilized small animal models. Considering that the epidermis and dermis of rodent animals are thinner and primarily healed via contraction instead of regeneration like in humans, it is necessary to validate the results in large animal models and, ultimately, clinical trials with long-term follow-ups [176]. The expenses and logistical obstacles associated with the industrial production of biomaterials for clinical applications continue to pose a significant barrier [176]. The proteolytic nature of the wound environment necessitates the careful selection of a dressing tailored to the specific wound type, whether acute, chronic, ulcerative, or resulting from burns. This characteristic also presents challenges for conducting blinded or randomized studies [53]. Lastly, the costliness of current advanced wound care products restricts their accessibility in hospitals and clinics [53,54]. Addressing this issue requires enhanced collaboration among industry stakeholders, researchers, and healthcare providers.

## 7. Conclusions

Medicine has made strides in understanding the function of fibroblasts in wound healing, though this field remains in its infancy in translating that knowledge to the clinical environment. Traditional medicines and cultures worldwide have employed plant and animal-derived compounds with fibroblast activity, laying the groundwork for future therapies. Modern scientific approaches have allowed for a more thorough understanding of how these natural substances result in physical, chemical, and cellular signals that modify fibroblast behavior. Biomimetic materials and cutting-edge delivery systems harness these effects and allow for an interplay between therapeutic delivery and the dynamic wound microenvironment. Utilizing compounds both ancient and new, harnessing the power of fibroblast activity allows for therapies promoting rapid and scar-free wound healing. 

## Figures and Tables

**Figure 1 ijms-25-03274-f001:**
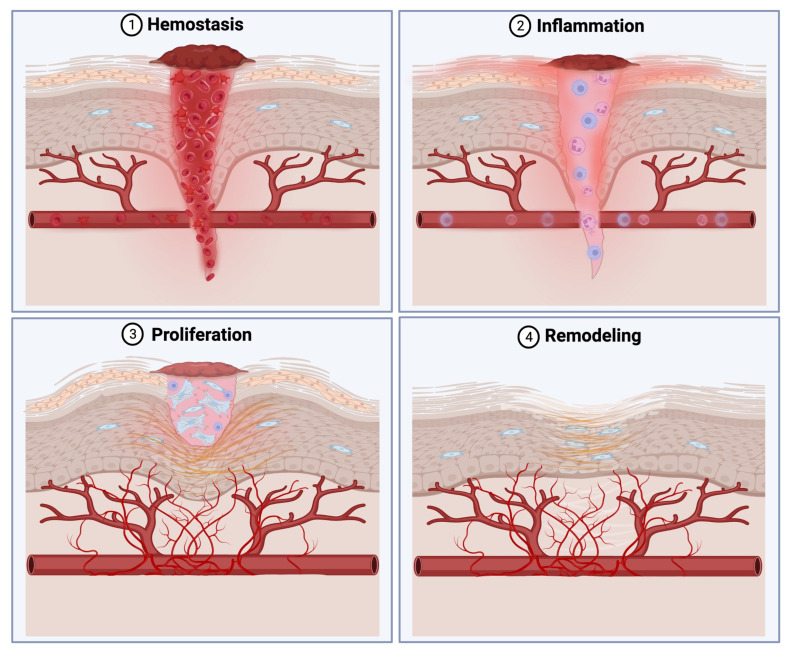
Summary of wound healing stages and fibroblast involvement.

**Figure 2 ijms-25-03274-f002:**
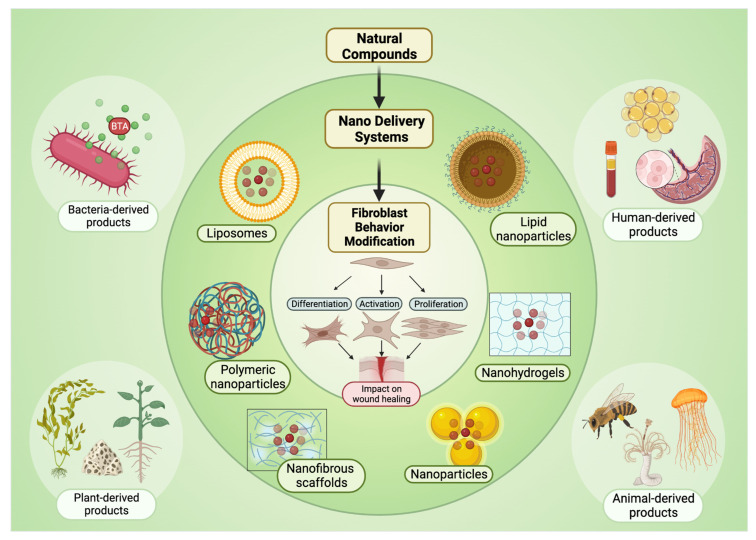
Summary of compounds/delivery systems and their influence on fibroblast behavior.

**Figure 3 ijms-25-03274-f003:**
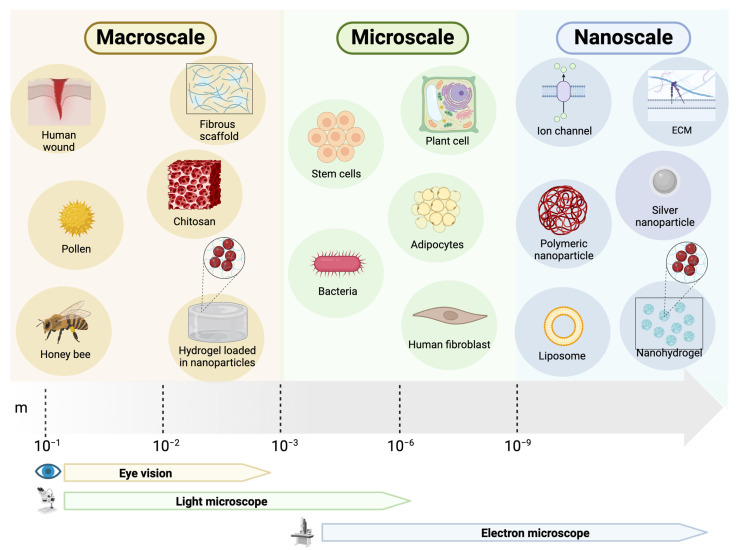
Relative scale of natural compound sources and delivery systems.

**Table 1 ijms-25-03274-t001:** Natural compounds.

Compound Category	Compound Name	Improvement of Wound Healing	Modification of Fibroblast Behavior	Sources
Plant-derived products	Sarıçiçek (*Achillea biebersteinni Afan*)	Anti-microbial, anti-oxidant, and anti-inflammatory properties	Downregulates TGF-β1 and upregulates bFGF expression at the gene and protein level in murine embryonic fibroblasts	Hormozi, 2019 [50] Mssillou, 2022 [45]
	*Astragalus propinquus*	Promotes re-epithelialization, revascularization, and immune function	Promotes cytokine secretion of TGF-β1, bFGF and EGF; promotes re-epithelialization; promotes proliferation, migration, and cell cycle progression of human skin fibroblasts	El-Ashram, 2021 [51]
	*Astragali Radix* and *Rehmanniae Radix*	Pro-angiogenic and anti-inflammatory properties	Activate genes in TGF-β1 pathway, regulate gene transcription for ECM synthesis via Smad pathway and cell motility via Ras/MAPK (non-Smad) pathway, and enhance skin fibroblast migration	Zhang, 2012 [52]
	Quercetin	Anti-bacterial, pro-angiogenic, and anti-oxidant properties	Modulates fibroblast activity, upregulate TGF-β1	El-Sherbeni, 2023 [53] Mssillou, 2022 [45] Falbo, 2023 [31]
	Curcumin	Anti-bacterial, anti-oxidant, anti-inflammatory, and pro-angiogenic properties.	Induces fibroblast proliferation and collagen deposition	El-Ashram, 2021 [51] Mssillou, 2022 [45] Falbo, 2023 [31]
	Luteolin	Anti-bacterial, anti-oxidant, anti-inflammatory, and pro-angiogenic properties	Modulates IGF, PDGF, and FGF	Mssillou, 2022 [45] Falbo, 2023 [31] El-Sherbeni, 2023 [53]
	Kaempferol	Anti-neoplastic, anti-inflammatory, anti-bacterial, and anti-oxidant properties	Increases hydroxyproline and collagen in wound tissue	Mssillou, 2022 [45] El-Sherbeni, 2023 [53]
	Icariin	Anti-oxidant, anti-inflammatory, and anti-apoptotic properties	Accelerates collagen deposition	Singh, 2019 [54] Mssillou, 2022 [45]
	Morin	Anti-oxidant, anti-inflammatory, and anti-apoptotic properties	Accelerates collagen synthesis	Ponrasu, 2018 [42] Mssillou, 2022 [45]
	Naringin	Anti-oxidant, anti-inflammatory, anti-microbial, and astringent properties	Accelerates collagen synthesis	Mssillou, 2022 [45]
	Catechin and Epigallocatechin-3-Gallate (EGCG)	Anti-bacterial, anti-oxidant, anti-inflammatory, and pro-angiogenic properties	Enhance wound contraction and modulate growth factors	Hernandez-Hernandez, 2017 [43] Mssillou, 2022 [45] Falbo, 2023 [31]
	Silymarin	Anti-oxidant and anti-inflammatory properties	Increases number of fibrocytes, improves alignment of healing tissues, and enhances collagen fibers and fibroblasts	Oryan, 2012 [55] Mssillou, 2022 [45]
	Hesperidin	Anti-inflammatory, anti-microbial, anti-fungal, anti-oxidant, anti-neoplastic, anti-hypertensive, pro-angiogenic, and anti-atherogenic properties	Upregulates TGF-β and Smad 2/3 mRNA expression	Li, 2018 [56] Mssillou, 2022 [45]
	Vicenin-2	Anti-oxidant, anti-inflammatory, and pro-angiogenic properties	Induces TGF-β to enhance fibroblast proliferation, migration and wound contraction	Tan, 2019 [43]
	Tannins	Anti-oxidant, pro-angiogenic, and antibacterial properties	Improves fibroblast proliferation and promote wound contraction	Li, 2011 [57] Falbo, 2023 [31]
	Terpinolene and α-phellandrene/α-pinene (PIN) and α-phellandrene	Anti-oxidant, anti-bacterial, anti-fungal, and anti-inflammatory properties	Improve migration and proliferation of fibroblasts	Bonnard, 2022 [58] Salas-Oropeza, 2020 [59] Salas-Oropeza, 2021 [60] Falbo, 2023 [31]
	Thymol	Anti-oxidant, anti-inflammatory, cicazitrant, anti-septic, anti-bacterial, and anti-fungal properties	Induces denser, thick, and parallel-arranged collagen fibers	Marchese, 2016 [61] Riella, 2012 [62] Falbo, 2023 [31]
	Taspine	Anti-bacterial, anti-inflammatory, anti-viral, and anti-neoplastic properties	Stimulates fibroblast chemotaxis, and induces hydroxyproline and KGF	Porras-Reyes, 1993 [63] Wang, 2022 [64] Vaisberg 1989 [65] Falbo, 2023 [31]
	Thymoquinone	Anti-microbial, anti-inflammatory, anti-oxidant, and anti-neoplastic properties	Enhances fibroblast formation and augments wound contraction	Algahtani, 2021 [66] El-Sherbeni, 2023 [53]
	APS2-1 from *Atragalus*	Anti-inflammatory, anti-oxidative, and immune-regulatory properties	Promotes fibroblast propagation and accelerate cell cycle progression, and promotes expression of TGF-β1, bFGF, and EGF	Zhao, 2017 [67] Yang, 2024 [68]
	ZWP from *Curcuma zedoaria*	Pro-angiogenic properties	Enhance collagen synthesis and deposition	Xu, 2018 [69] El-Sherbeni, 2023 [53]
	Extract of *Sargasum Ilicifolium* seaweed species	Increased speed of wound closure	Increases myofibroblast activity, promote TGF-β1 expression	Premarathna, 2020 [49]
	Madecassoside and Asiaticoside from *C. asiatica*	Improved speed and quality of wound healing	Activate the TGF-β/Smad pathway, enhancing type I and III collagen expression	Wu, 2012 [70]
Animal-derived products	Honey	Anti-bacterial, anti-oxidant, anti-inflammatory, and pro-angiogenic properties	Induces fibroblast proliferation and migration, and induces collagen matrix development	Ratcliffe, 2014 [71] Ibrahim, 2018 [72] El-Ashram, 2021 [51]
	Sericin	Anti-oxidant, pro-angiogenic, and anti-inflammatory properties	Regulates TGF-β1 and TGF-β3 expression, and activates collagen production	El-Ashram, 2021 [51]
	Maggot excretions/secretions of *Phaenicia sericata*	Anti-inflammatory, pro-angiogenic, anti-viral, and anti-neoplastic properties	Activate and enhance the growth rate of fibroblasts	Prete, 1997 [73] Ratcliffe, 2014 [71]
	Marine collagen	Pro-angiogenic, and anti-aging properties	Promote fibroblast migration	Geahchan, 2022 [33] Chandika, 2015 [74]
	Sea cucumbers	Anti-bacterial, anti-inflammatory, anti-oxidant, and immune regulatory properties	Stimulate fibroblast chemotaxis and proliferation, and breakdown of ECM proteins	El-Ashram, 2021 [51] Ibrahim, 2018 [72]
Exogenous growth factors	TGF-β	Pro-angiogenic, and immune regulatory properties	Prompts differentiation of fibroblasts into myofibroblasts and ECM formation/ deposition via TGF-β1 and TGF-β2 chemotaxis of fibroblasts via TGF-β3	Walraven, 2017 [75] Barrientos, 2008 [76] Dolati, 2020 [77]
	EGF and NRG1	Anti-inflammatory and pro-angiogenic properties, and enhanced kerotinocyte recruitment and cell motility	Stimulate recruitment of fibroblasts	Yoon, 2018 [78] Dolati, 2020 [77]
	FGF-2 (bFGF)	Anti-inflammatory and pro-angiogenic, properties and enhanced kerotinocyte recruitment and wound contraction	Modulates ECM formation and inhibits TGF-β1/Smad-dependent pathways	Borena, 2015 [79] Dolati, 2020 [77]
	IGF-1	Anti-inflammatory, anti-apoptotic, and pro-angiogenic properties, and stimulator of keratinocyte proliferation	Stimulates proliferation of fibroblasts and collagen	Todorović, 2008 [80] Dolati, 2020 [77]
	hPDGF and rPDGF-B	Pro-angiogenic properties; stimulates chemotaxis of polymorphonuclear leukocytes and monocytes; induce MMPs and tissue inhibitors of metalloproteinases (TIMPs)	Stimulate fibroblast mitogenesis and chemotaxis; promote procollagen type I synthesis	Pierce, 1988 [81] Dolati, 2020 [77]
Growth Factor-rich products	Platelet-rich plasma (PRP)	Anti-microbial, anti-inflammatory, and pro-angiogenic properties	Consists of high levels of PDGF and TGF-β1	Fotouhi, 2018 [82] Dolati, 2020 [77]
	Platelet-rich fibrin matrix (PRFM)	Anti-inflammatory and pro-angiogenic properties	Stimulates release of PDGF, TGF-β1, EGF, FGF-2, and IGF	Lin, 2018 [83] Dolati, 2020 [77]
	Decellularized and dehydrated human amniotic membrane (DDHAM)	Anti-bacterial, anti-inflammatory, and pro-angiogenic properties	Stimulate release of PDGF, TGF-β1, EGF, FGF-2, TGF-a, placental GF, G-CSF, Interleukin (IL)-4, IL-10, and various TIMPs	Sheikh, 2014 [84] Smiell, 2015 [85] Dolati, 2020 [77]

**Table 2 ijms-25-03274-t002:** Delivery systems.

Nano DDS	Natural Compound	Modification of Fibroblast Behavior	Sources
Liposomes Ability to improve bioavailability, cause sustained transdermal delivery of different medicinal compounds, and overcome possible drug overdose and toxicity.	Curcumin	Shortens inflammatory process, inhibits bacterial growth, promotes fibrosis, angiogenesis, re-epithelialization, and wound contraction	Kianvash, 2017 [131]
	Madecassoside	Significant burn wound healing effect	Li, 2016 [132]
	Usnic acid	Inhibits the secretion of pro-inflammatory cytokines, TNF-α, IL-6, IL-1B Induces nitric oxide and cyclooxygenase-2 (COX-2) Increases IL-10 and HO-1 in a dose-dependent relation Anti-bacterial activity Enhances maturation of granulation tissue and better collagen deposition	Nunes, 2016 [133]
	bFGF	Promotes fibroblast proliferation, migration, differentiation Expedites regeneration of vascular vessels and the synthesis of procollagen and collagen matrix	Xu, 2017 [134]
	Insulin/chitosan	Increases re-epithelialization collagen content, granulation tissue, wound tensile strength, and local production of insulin-like growth factors by fibroblasts. Increases proliferation and migration of human keratinocytes, which stimulates cell growth and enhances wound healing	Dawoud, 2019 [135]
Transfersomes Deformable liposomes with an edge activator.	Gellan cholesterol nanohydrogels/baicalin	Inhibits TNF-αInhibits IL-1β Visually improves wound healing	Manconi, 2018 [136]
Polymeric nanoparticles Protect the degradation of drugs and release them in a controlled manner.	PDGF-A, IGF-1, EGF	Advanced granulation tissue formation, significantly enhances healing of chronic wounds	Choi, 2017 [137]
	PLGA/LL-37	Increases collagen deposition, and organization, enhancement of epithelialization, and neovascularization	Chereddy, 2014 [138]
	hVEGF gene/stem cells	Enhances angiogenesis, and reduces tissue degeneration and fibrosis in ischemic limbs	Yang, 2010 [136]
	Thymol/chitosan/AgNPs	Excellent anti-bacterial properties	Manukumar, 2017 [139]
Inorganic nanoparticles Deprived from inorganic materials, including metallic nanoparticles, carbon-based nanoparticles, and ceramic nanoparticles. Benefiting from the intrinsic nature of materials, inorganic nanoparticles exhibit both similar merits in wound healing treatment and a strong anti-bacterial effect.	Iron oxide/thrombin	Increases tensile strength of wounds, decreases inflammation	Ziv-Polat, 2010 [140]
	Cerium oxide	Enhances fibroblast proliferation, myofibroblast differentiation Accelerates migration and tube-forming ability of vascular endothelial cells	Chigurupati, 2013 [141]
	Zn02	Shows anti-bacterial activity and enhances wound healing	Ali, 2017 [142]
	Gold	Enhances of wound healing, increases collagen expression, decreases MMP-1 expression and TGF-B1 Enhances VEGF, angiopoietin 1, and 2	Kim, 2015 [143]
	Silane/amphotericin B	Shows efficacy in controlling Candida infection	Sanchez, 2014 [144]
Lipid nanoparticles Introduced to overcome the limitation of liposomes. Controlled release of drugs due to their nontoxic colloidal dimensions.	rhEGF	Enhances proliferation and migration of fibroblasts, wound contraction, and epidermal regeneration	Gainza, 2013 [145]
Nanofibrous structures Mimic the ECM, provide favorable conditions for cell attachment and contact with drugs. Enhance variety of therapeutics agents due to their high area-to-volume ratio.	Andrographolide/silica	Accelerates wound healing, increases collagen deposition in the wound site, decreases inflammation	Jia, 2018 [146]
	Astragaloside IV	Accelerates wound healing and inhibits scar formation, increasing angiogenesis, regulating newly formed types of collagen, and improving collagen organization	Shan, 2015 [147]
	PDGF-BB/VEGF	Accelerates wound healing, promotes fibroblast growth and inhibits bacteria in vitro	Xie, 2013 [148]
	Lawsone	Significantly increases TGF-β1 and collagen gene expression in vitro and promotes re-epithelialization of the wound in vivo	Abadehie, 2021 [149]
Nanohydrogel High flexibility, high hydrophilicity, high mechanical strength, tunable structure, and the ability to absorb wound exudates as well as permeate oxygen and prevent wound dehydration.	Cellulose nanocrystal and hyaluronic acid/chitosan NPs/GM-CSF	Enhances proliferation and differentiation of fibroblasts, lowers inflammation, and increases collagen deposition	Dehkordi, 2019 [150]
	Carrageenan/nano silicates	Enhances cell adhesion and spreading, reduces blood clotting time, facilitates in vitro tissue regeneration and wound healing	Lokhande, 2018 [151]
	Nanocellulose/acrylic acid hydrogels	Maintains the activity and morphology of human dermal fibroblasts, promotes rapid cell proliferation, and affects 9 genes’ expressions related to wound healing	Loh, 2018 [152]
Hydrogels loaded with nanoparticles Synergistic effect between hydrogels and nanoparticles encapsulated.	Chitosan hydrogels/phenytoin	Increases the content of collagen fibers and fibroblasts in the wound tissue	Cardoso, 2019 [153]
	Thermosensitive hydrogel/gold NPs	Enhances skin re-epithelialization, granulation tissue, vascularization, and collagen deposition Modulates gene expression of inflammatory and anti-inflammatory mediators	Mahmoud, 2019 [154]
	Hydrogels/cyclosporine A solid lipid NPs	Significantly increases rate of mucosal repair	Karavana, 2012 [155]
	Hyaluronic acid and chondroitin sulfate/asiatic acid/ZnO NPs/CuO NPs	Raises DNA, total protein, hexosamine and hydroxyproline content, and leads to superior re-epithelization, collagen fiber arrangement and angiogenesis	Thanusha 2018 [156]
	Hydrogel/Simvastatin polymeric NPs	Enhances of epithelialization and wound healing Decreases inflammatory cell infiltration	Aly, 2019 [157]
	PRP/collagen NPs	Enhances epithelialization and wound closure	Shalaby, 2023 [158]

## Data Availability

Not applicable.

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
