# Peer review of "Natural Compounds and Biomimetic Engineering to Influence Fibroblast Behavior in Wound Healing"

_ijms, 2024, doi:10.3390/ijms25063274_

Round 1

Reviewer 1 Report

Comments and Suggestions for Authors

This literature review discusses the role of natural compounds and biomimetic engineering in modulating fibroblast behavior to enhance wound healing. It outlines the historical use of natural products in wound management and examines how these substances, alongside advanced biomimetic materials and delivery methods, affect fibroblast activity to improve healing outcomes. The review addresses the stages of wound healing, the impact of various natural and synthetic compounds on fibroblast functions, and the development of new delivery systems aimed at optimizing therapeutic effects in wound care. It concludes by emphasizing the potential of combining traditional remedies with modern scientific approaches for effective and scar-free healing. I would like to make my amendments and suggestions in this paragraph

While acknowledging the positive aspects of this literature review, it is important to point out areas for improvement to enhance the quality of the article. Fibroblasts play a crucial role in scar formation and fibrosis, making them a focal point in research to improve wound healing outcomes. These cells, abundantly present in wounds, coordinate the healing process at every stage by producing regulatory molecules and recruiting immune cells. Initially, wound healing involves coagulation and inflammation in response to damaged extracellular matrix and resident cells. The dysfunction of fibroblasts could lead to delayed wound closure or excessive fibrosis. There is significant heterogeneity in dermal fibroblast responses to Wnt/β-catenin signaling(HanFront. Cell Dev. Biol2022)(Hamburg-Shields et al., 2015)(Mastroianni et al., 2016(Rognoni et al., 2016)(Lim et al., 2018(Phan et al., 2020), which has been previously attributed to the effects of wound macrophages on hair follicle stem cell activationWang et al., Nature Communication2017)(Rahmani et al., 2020)(Chen et al,PLOS ONE,2017)(Chen et al,Theranostics,2019.

The study notes the importance of fibroblasts in the proliferative phase of wound healing and their role in remodeling the extracellular matrix, which can lead to complications if dysregulated. It also touches on the development of synthetic devices for wound healing and scar management, as well as the interest in natural compounds for their regenerative capabilities. Understanding how these compounds affect wound repair and fibroblast modulation is considered crucial.……. In response to the demonstrated relationship between mechanical force and fibroblast behavior, various synthetic external tension off-loading devices and dressings have been developed for wound healing and scar management.9 As cellular pathways associated with mechanotransduction have been identified, new avenues have been opened for targets of pharmaceutical drug development. Future therapies may target sites of mechanotransduction, like mechanosensitive calcium-dependent channels, Rho-GTPases, and the ILK-PI3K/Akt pathway which are activated by stretch and integrin stress. Similarly, focal adhesion kinase (FAK), a non-receptor tyrosine kinase, is phosphorylated upon mechanical injury; initiating inflammatory signaling, fibroblast recruitment, and collagen production. Downstream pathways, such as those associated with wnt, AKT/β-catenin, and yes-associated protein (YAP), alter transcription and initiate profibrotic processesWang et al., Nature Communication2017(Chen et al,PLOS ONE,2017)(Chen et al,Theranostics,2019.

Comments on the Quality of English Language

Minor editing of English language required

Author Response

Reviewer 1, the Authors thank you for your time and careful consideration of our manuscript. We are pleased you have found positive aspects in our literature review. Thank you, as well, for your suggestions to improve our manuscript, which we agree have enhanced the quality of our draft. In response to your comments, we have adopted your suggested phrasing and made substantial revisions to our introduction. Specifically, we have elaborated on the role of fibroblasts in wound healing, the heterogeneity in response to wnt/β-catenin signaling, and mechanotransducive pathways. We have also made the addition of your suggested citations to our paper to incorporate the role of wnt signaling discovered through study of hair follicle regeneration.

Reviewer 2 Report

Comments and Suggestions for Authors

Major comments:

1.    “Introduction” (page 2): Here, the authors described the process of wound healing in general and the role of fibroblast in this process. However, the (rather important) role of other cell types (including endothelial cells and immune cells) as well as of major mediators (such as factors of the coagulation cascade, Tissue factor (F3) and so on) should also be described in brief in this context (as done later in the text for other proteins involved in signalling (e.g. see Akt/PI3K, chemokines and receptors etc.).

2.    “2. Natural Compounds” (page 3 and 4): In this part of the manuscript, the authors mentioned some plants/species utilized in natural medicine. In this context, please also describe which factors/substances (in these plants) are suggested to be the initiators of the described effects, e.g. in Achillea biebersteinii Afan. This also applies for the species mentioned below (in the text).

Comments on the Quality of English Language

Grammar, style and phrasing (English style and editing): Sometimes space characters are missing or too much in the text (e.g., see table 1 and check the whole manuscript). Use a uniform style for words including the term “anti” with or without hyphen (e. g. see page 9 “antimicrobial” vs “anti-biofilm”, page 12 “anti-scarring” and “anti-scar” vs “antibacterial”, or Table 1 “Antibacterial” vs “Anti-neoplastic, anti-inflammatory”, “Anti-oxidant” and so on. Moreover, please begin entries in table 1 consistently with small or capital letters “Antimicrobrial, anti-inflammatory, anti-oxidant, and anti-neoplastic properties” vs “anti-inflammatory, anti-oxidative, and immune regulatory properties”. Please define all abbreviations used in the manuscript (e. g. in page 2: “CXCL8”, “ILK-PI3K/Akt”, page 5: “IGF-1, EGF, and NRG1”, page 8: “NPs” (defined some lines post mentioned for the first time), “TNF-alpha”, page 9: “α-SMA”, page 11: “HLA”, “GM-CSF”, SNPs“, “GelMA+ β-cd UV”, page 12: “AI”, Table 1: “SMAD”, “Smad”, “TIMPs”, “TNF-alpha, IL-6, IL-1B” etc.). Use a uniform style for the term “wound healing” with or without hyphen (e.g. see Abstract). Please use a uniform style for the term “et al” and “in vitro”/”in vivo” in italics or not (e.g. see page 9: “Gainza et al.” vs "Chereddy et al. introduced" or “Ziv Polat et al.” vs “Kim et al.”, page 10: “in vitro” ”in vivo” alternating between cursive or not). And use a uniform style for the term “SMAD” or “Smad” in table 1.

Author Response

Reviewer 2, the Authors thank you for your time and careful consideration of our manuscript. Thank you, as well, for your suggestions to improve our manuscript. We have made the following changes based on your suggestions:

  1. In the introduction, we expanded information regarding the cellular and platelet contributions to various stages of wound healing. As suggested, we focused on the inclusion of information related to immune cells and the coagulation cascade. We added the following several sentences in the introduction of the manuscript:
    1. Page 2: “Fibroblasts play a crucial role in scar formation and fibrosis, making them a focal point in research to improve wound healing outcomes.4 These cells, abundantly present in wounds, coordinate the healing process at every stage by producing regulatory molecules and recruiting immune cells.5 The dysfunction of fibroblasts could lead to delayed wound closure or excessive fibrosis. Initially, wound healing involves coagulation and inflammation in response to damaged extracellular matrix (ECM) and resident cells. This first inflammatory phase includes initial vasoconstriction of injured vessels and a complex sequence of reactions to blood contact involving binding of plasma factor VII/VIIa to tissue factor 3 to ultimately lead to the production of a clot formed by platelets and fibrin.”
    2. Page 2: “Platelets also participate in increasing the permeability of blood vessels mediated by histamine and serotonin secretion as well as endothelial cells that produce nitric oxide to allow vasodilation. Signaled by platelet products, complement activation, and coagulation cascade factors, chemotaxis leads to immune cell migration to the wound bed to eliminate potential pathogenic or foreign debris. Although on one side, neutrophils are not considered critical to wound healing, they produce inflammatory products to induce wound healing and are responsible for phagocytosing pathogens. Skin resident as well as monocyte-derived macrophages, on the other side, play a critical role in wound healing and become the dominant cell population to orchestrate growth factors in the wound environment until fibroblast proliferation (30524911). Fibroblasts, typically arriving at the wound site around day 5-7, as well as lymphocytes, release inflammatory cytokines and chemokines, thus modulating the wound microenvironment.6”
  2. Factors suggested as initiators for the described effects on wound healing of Achillea biebersteinii Afan, Centella asiatica, Astragali Radix, and Rehmanniae Radix have been included on page 3 and 4 the text
  3. Erroneous extra spaces have been removed from the tables and text
  4. Uniform style for words including the “anti-“ prefix has been implemented throughout text and tables
  5. Uniform capitalization has been implemented in table entries
  6. All abbreviations have been defined and uniform formatting has been applied. Abbreviations not previously included have been added to the glossary at the end of the text. Abbreviations have been applied to the first mention of the term in the text.
  7. “Wound healing” has been uniformly formatted throughout the text
  8. “Et al.” has been uniformly formatted throughout the text
  9. In vivo” and “In vitro” have been uniformly formatted throughout the text
  10. “Smad” has been uniformly formatted throughout the text

Reviewer 3 Report

Comments and Suggestions for Authors

The manuscript explores the pivotal role of natural compounds and biomimetic engineering in influencing fibroblast behavior for effective wound healing. It delves into the historical context, identifying specific natural compounds with wound-healing properties, and highlights modern scientific techniques and delivery methods. The review concludes by outlining innovative future strategies, emphasizing the integration of ancient remedies with contemporary approaches while acknowledging challenges in translating these advancements to clinical settings. The topic has broader reach to researchers and manuscript is written well, however, there are many areas which needs to be improved. My comments are appended below-

  1. Introduction Enhancement and knowledge gap clarity: Clearly state the primary objective of the manuscript at the beginning to guide readers through the focus on natural compounds, biomimetic engineering, and their influence on fibroblast behavior in wound healing. Expand the introduction to include a brief historical perspective on the use of natural products in wound healing, highlighting key milestones in the field to provide context for readers. In addition, on page 2, cite a recent report https://doi.org/10.1080/10496475.2023.2267467 with the sentence ´Understanding how these compounds can affect wound repair and modulate fibroblast behavior is of significant interest´ to make references up to date.
  2. Quantification of Impact and Terminology Consistency: Provide quantitative data on the historical and current impact of natural compounds on wound healing to emphasize the significance of the subject matter. Ensure consistency in the use of terminology related to fibroblast behavior and wound healing throughout the manuscript to avoid confusion.
  3. Citation of Modern Scientific Techniques Highlighting Identified Compounds: Explicitly mention specific modern scientific techniques that have contributed to the detailed understanding of how natural compounds influence fibroblast behavior in wound healing. Include a section explicitly listing and discussing specific natural compounds known for their wound-healing properties, providing details on their mechanisms of action.
  4. Explain Biomimetic Engineering Principles: Elaborate on the principles of biomimetic engineering and provide examples of materials and devices developed through this approach to create a better understanding for readers.
  5. In-depth Discussion on Delivery Methods: Expand the discussion on micro- and nano-scale delivery methods, detailing advancements and challenges associated with delivering natural and biomimetic compounds through the stratum corneum.
  6. Critically Analyze Biomimetic Compounds: Provide a critical analysis of biomimetic compounds designed to replicate the natural wound microenvironment, discussing their successes and potential limitations. Cite a latest study along with the reference 109 cited therein https://doi.org/10.1002/cmdc.202300328 with the sentence ´ Nano-particles of natural origin are created to address shortcomings with current drug delivery methods, such as limited duration of action, short half-life, and adverse reactions ´
  7. Thorough Examination of Future Strategies: Elaborate on each of the mentioned innovative future strategies, such as fibroblast-influencing pathways, responsive biomaterials, smart dressings, real-time monitoring, and stem cell applications, providing in-depth insights.
  8. Discussion on Translation Challenges & Integration of Ancient Remedies Published: Address the challenges mentioned in translating findings to clinical settings, specifically delving into the limited validation of biomaterials in large animal models and logistical obstacles in industrial production. Offer a more detailed discussion on the integration of ancient remedies with modern approaches, explaining how this integration holds promise for achieving effective and scar-free wound healing.

Comments on the Quality of English Language

Minor editing of English language required

Author Response

Reviewer 3, the Authors thank you for your time and careful consideration of our manuscript. We are pleased that you found our manuscript well-written and has the potential for broad reach. Thank you, as well, for your suggestions to improve our manuscript. In response to your feedback, we have made significant revisions to our manuscript to increase clarity, specificity, and critical analysis. We feel these changes have resulted in substantial improvement to our work, and have summarized them as follows:

  1. A statement of the paper’s intent has been added to the introduction (Page 2) and reads: “This review aims to highlight natural compounds capable of modulating fibroblast behavior, consequently facilitating wound regeneration and repair. Additionally, it delves into the realm of biomimetic engineering, where researchers draw inspiration from nature to design materials and devices that replicate physiological cues essential for optimal wound repair.”
  2. The recommended citation has been added after the sentence “Understanding how these compounds can affect wound repair and modulate fibroblast behavior is of significant interest.”
  3. The introduction has been expanded with historical context and landmarks added to provide context for the reader.
  4. The introduction to Section 2.1 (Page 4) has been expanded to include quantitative data to emphasize the significance of the subject matter.
  5. The introduction (paragraph on Page 3) has been expanded to include relevant modern scientific techniques.
  6. Specific natural compounds known for their wound-healing properties and details of their mechanism of action have been expanded upon in the last paragraph of Section 2.1. (Page 4)
  7. Terminology related to fibroblast behavior and wound healing has been standardized to maximize ease of interpretation for readers throughout the manuscript.
  8. In section 3: “Biomimetic Engineering of Biomaterials” we have added a description of biomimetic engineering, including examples intended to provide a general understanding of the field, as well as specific technologies relevant to tissue regeneration and wound healing. This paragraph introduces section 3 on page 6, beginning with “Biomimetic engineering involves…”.
  9. In section 3, we have elaborated on the polymers used to engineer biomimetic dressings, including the most important parameters in polymer selection, as well as the most important advantages and disadvantages of polymers relevant to our discussion. These updates can be found in the paragraph beginning on page 6, which reads “Electropsun nanofibers are the preferred method…”.
  10. We have updated the citation of the sentence “Nanoparticles of natural origin are created to address shortcomings with current drug delivery methods, such as limited duration of action, short half-life, and adverse reactions” on page 11. We have also rephrased the sentence slightly to more accurately reflect conclusions drawn in the literature. The sentence now reads: “Natural NPs exhibit improved duration of action and biocompatibility compared to other nanoparticles or pure drugs”.
  11. We have expanded the discussion of micro and nanoscale delivery systems, focusing on delivery challenges to the stratum cornea. We have also added text highlighting advancements in the field. In light of your suggestions to improve our work, we have modified the discussion on delivery methods as follows:
    1. Page 7: “Indeed, the journey of topical compounds to penetrate the wound site is limited by several physiological barriers, and passive drug diffusion into the skin is generally impeded by the outermost lipophilic layer of the skin, the stratum corneum. This tight layer is composed of lipids including triglycerides, cholesterol, and free fatty acids that are linked by covalent bonds with corneocytes and provide a defensive barrier to the external environment. Furthermore, only moderately lipophilic compounds can perform transcutaneous passage. To address these limitations, both natural and synthetic biomaterials have been developed to fuse with stratum corneum, disrupt the skin surface, or accumulate in the hair follicles to act as a long-term reservoir. These biomaterials also must be delivered in a controlled manner to obtain effective concentrations of therapeutics drugs as well as targeting specific cells such as fibroblasts, and cell components. to obtain a stronger local effect while decreasing potential side effects, (Table 2. Delivery systems). However, enhancement of transdermal drug delivery could be achieved using penetration enhancers, disruption of the skin, or physical techniques, innovative formulation designs using new technologies have emerged as an effective, reliable method for safe delivery of natural compounds.”
    2. Page 7: “Rapidly, new therapeutic alternatives using engineering have been developed in the micro and nanoscale. In addition to the size difference, there is a range of physical and chemical properties that differ between those particles such as melting point, dissolution, and solubility, allowing them to be used in different indications.”
  12. The Authors have added an entirely new section titled “Micro delivery systems” (Page 7) discussing the challenges and opportunities offered by the field. We have explicitly discussed here examples of how this technology can be used to aid in the delivery of natural compounds.
  13. In the Future Strategies section, we have expanded more information on responsive biomaterials, smart dressings, real-time monitoring, and stem cell applications. We have added more examples of each material along with their strengths, limitations, mechanisms of action, and related fibroblast responses when applicable (Pages 12-13).
  14. In the Limitations section, we added limitations about nanofiber-based substrates and challenges in validating biomaterials/smart dressings in large animal models and industrial production. The relevant section now reads:
    1. “[M]any experiments conducted in biomaterials application in wound healing have utilized small animal models. Considering that the epidermis and dermis of rodent animals are thinner and primarily healed by contraction instead of regeneration like in humans, it is necessary to validate the results in large animal models and, ultimately, clinical trials with long-term follow-up. Firstly, the expenses and logistical obstacles associated with the industrial production of biomaterials for clinical applications continue to pose a significant barrier. Firstly, the proteolytic nature of the wound environment requires the appropriate selection of a specific dressing to fit the wound type (acute, chronic, ulcer, burns, etc). This also makes it more difficult to conduct blinded or randomized studies. Lastly, current new wound care products are expensive, limiting their use in hospitals and clinics. To address this issue, there needs to be increased collaboration between the industry market segments, researchers, and healthcare providers.”

Reviewer 4 Report

Comments and Suggestions for Authors

The manuscript titled "Natural Compounds and Biomimetic Engineering to Influence Fibroblast Behavior in Wound Healing" has been evaluated. The review provides a comprehensive summary of the effectiveness of natural compounds in enhancing fibroblast behavior, thereby promoting wound regeneration and repair. Additionally, it explores the field of biomimetic engineering, wherein researchers derive inspiration from nature to create materials and devices that mimic physiological cues, which are crucial for effective wound healing. The article concludes by describing the latest delivery mechanisms that aim to improve the bioavailability of natural compounds. It also explores innovative future strategies, such as exploring fibroblast-influencing pathways, responsive biomaterials, smart dressings with real-time monitoring, and stem cell applications. However, several challenges, such as the limited validation of biomaterials in large animal models and logistical obstacles in industrial production, need to be addressed before translating these findings to clinical settings. The integration of ancient remedies with modern approaches holds promise for achieving effective and scar-free wound healing, but the study requires significant revision before it can be considered for publication.

Several formatting issues, such as the lack of mention of authors' educational qualifications after their names, and the incorrect numbering of references, which begin with 2, 3 instead of 1, have been identified. The reference citation is also confusing. Furthermore, Table 1 is not cited in the main text, and the caption is missing in Table 2, which is also not cited. To improve the manuscript, it is recommended that a figure be added to illustrate the role of fibroblast behavior in wound healing.

In conclusion, the manuscript provides valuable insights into the use of natural compounds and biomimetic engineering to enhance wound healing. However, to ensure its suitability for publication, it requires significant revision to address the formatting and citation issues, add a figure, and improve its overall readability.

Comments on the Quality of English Language

Minor editing of English language required

Author Response

Reviewer 4, the Authors thank you for your time and careful consideration of our manuscript. We are pleased that you found our manuscript to be comprehensive and provide valuable insights. Thank you, as well, for your suggestions to improve our manuscript. We have made the following changes based on your suggestions:

  1. The author’s qualifications (degrees earned) have been listed after their names
  2. The references have been corrected to begin with 1
  3. Table 2 has been given a caption and in-text references have been added referring to both Table 1: Natural compounds and Table 2: Delivery systems.
  4. A new figure (Figure 1) has been added overviewing the stages of wound healing and fibroblast contribution.
  5. In response to Reviewer 4 as well as others, substantial editing has occurred to improve the overall flow and readability of the manuscript.

Round 2

Reviewer 2 Report

Comments and Suggestions for Authors

w/o

Comments on the Quality of English Language

W/o

Reviewer 3 Report

Comments and Suggestions for Authors

accept

Reviewer 4 Report

Comments and Suggestions for Authors

The authors performed significant revisions based on the reviewer's comments, the MS can be acceptable for publication.